# EFFICIENT PARAMETER TUNING OF LARGE PROTEIN LANGUAGE MODELS FOR *De Novo* PROTEIN DESIGN

## ABSTRACT

Protein language models (ProtLMs) have achieved unprecedented breakthroughs in protein design. However, optimizing ProtLMs effectively with limited data has been challenging due to their large number of parameters. In this study, we introduce prefix tuning to efficiently prompt the pre-trained ProtLMs for *de novo* protein design with desired structures and functions. During the training process, only the prefix virtual token is trainable, while the pre-trained ProtLM is frozen. We trained two prefix virtual tokens on antimicrobial peptide (AMP) dataset and alpha-helix strucutre dataset, respectively. Our results demonstrate that prefix tuning is efficient to prompt the pre-trained ProtLM by optimizing fewer trainable parameters to achieve superior results compared with fine tuning, even under low-data settings. Furthermore, these two prefix virtual tokens can be combined to precisely control protein generation with both desired properties, which is not possessed by other tuning methods. We anticipate that prefix tuning will contribute to the protein discovery and biomedical advancement.

## 1 INTRODUCTION

Designing proteins with tailored functions and structures is a primary goal of protein engineering research. In the past decade years, protein engineering has achieved huge advancements through directed evolution (Romero & Arnold, 2009). Nonetheless, this technology heavily depends on random mutations and heuristic search methods, which can only delve into a minute fraction of the vast protein sequence space. Deep learning-based *de novo* protein design methods, such as deep generative models, typically generate proteins by sampling from the latent space of protein representations, which can explore the whole protein space for designing novel proteins.

Recently, language models, such as BERT(Devlin et al., 2018) and GPT(Brown et al., 2020) derived from natural language processing (NLP), have been introduced into the field of protein research. Several protein language models (ProtLMs), such as ESM-1b(Rives et al., 2021) and Prot-GPT2(Ferruz et al., 2022), have been developed and applied in substantial downstream tasks, such as protein function prediction and protein design. By leveraging the vast amount of available protein-related data, ProtLMs can predict the potential functions and interactions of proteins more accuratelyWang et al. (2023). This is particularly valuable in drug discovery, where understanding protein functions can aid in target identification and drug design. ProtLMs can also assist in designing novel proteins with specific structures or functions. By generating protein sequences that meet desired criteria, these models contribute to the development of new enzymes, therapeutics, and biomaterials. However, due to the huge number of parameters of ProtLMs, adapting ProtLMs to a specific domain or a particular task, especially under data-limited scenarios, can encounter some challenges, including overfitting and catastrophic forgetting.

Prompt learning (Gao et al., 2020) has been proven highly successful in numerous small data scenarios in NLP. For instance, large-scale language models like GPT-3 (Brown et al., 2020) can get remarkable few-shot results through natural language prompts, without tuning the models' parameters. Regrettably, the vocabulary of ProtLMs is confined to amino acids, preventing them from fully leveraging natural language prompts (Brown et al., 2020) (**Figure 1A**). Nevertheless, prefix tuning (Li & Liang, 2021) can train virtual tokens to prompt language models in downstream tasks, offering promising possibilities for advancements in protein design.

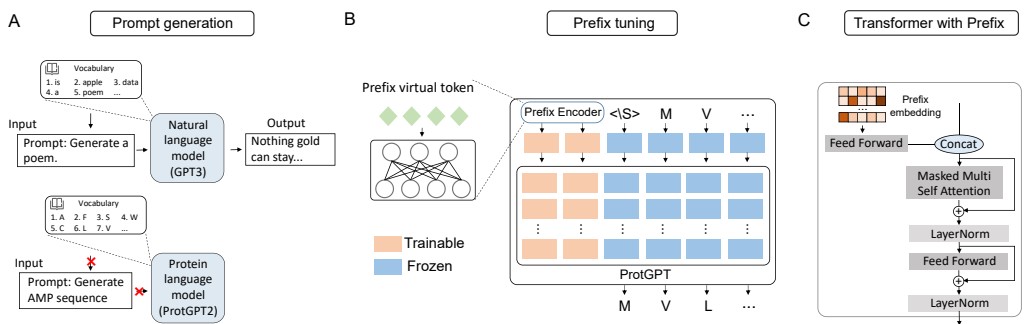

Figure 1: Illustration of the proposed method. (A) Prompt generation. The amino acid vocabulary can't be used to construct valid prompts of ProtLM. (B) Prefix-tuning architecture, where the pre-trained protein language model's parameters are frozen and only the prefix-tuning parameters are trainable. (C) Concatenation with prefix embedding.

In this study, we introduce prefix tuning to efficiently prompt the pre-trained ProtLM for protein design with desired structures and functions. Prefix tuning trains a virtual token to direct the pre-trained ProtLM in generating protein sequences. During the training process, only the prefix virtual token is trainable, while the pre-trained ProtLM is frozen. We assess the effectiveness of prefix tuning to direct ProtGPT2 in generating proteins with alpha-helix structure and antimicrobial function. In summary, our contributions include: (1) introducing prefix tuning to prompt the pre-trained ProtLM for protein design; (2) demonstrating the effectiveness of prefix tuning by optimizing fewer trainable parameters to achieve superior results compared with fine tuning, even under low-data settings; (3) combining two prefix virtual tokens to precisely control protein generation with both desired properties. Moreover, to the best of our knowledge, this is the first study that leverages prefix tuning to prompt ProtLMs for protein sequence generation. We aspire for this method to gain widespread adoption and contribute to the advancement of drug design and protein discovery.

## 2 RELATED WORK

**Rosetta-based protein design.** The majority of traditional protein design methods are based on Rosetta tool (Leaver-Fay et al., 2011; Leman et al., 2020; Huang et al., 2011), which has been effectively employed to address various protein design challenges, such as enzyme design (Richter et al., 2011), vaccine design (Correia et al., 2014), and protein drug design (Cao et al., 2022). The Rosetta software suite is a comprehensive tool for macro-molecular modeling, encompassing biomolecular structure prediction and design, enabling researchers to measure and manipulate protein conformations, calculate energies, model variable regions of proteins, dock proteins or small molecules, and design protein sequences through the Rosetta suit.

**Structure-based protein generation model.** A structure-based generation model is trained using structural information, such as atomic coordinates, contact maps, and inter-residue angles, to directly generate protein structures. For instance, IG-VAE (Eguchi et al., 2020) employed a variational auto-encoder (VAE) to directly generate the 3D coordinates of immunoglobulins. Generative Adversarial Networks (GANs) were also applied to generate protein structures encoded by pairwise distances between $\alpha$-carbons on the protein backbone (Anand et al., 2019). Recently, diffusion models have made significant strides in various generative tasks (Saharia et al., 2022). Diffusion models train neural networks starting from noise and generate samples through iterative denoising processes. They are also applied in protein structure generation (Luo et al., 2022; Watson et al., 2023).

**Sequence-based protein generation model.** Sequence-based generation methods can directly produce protein sequences with a desired function. (Hawkins-Hooker et al., 2021) introduced MSA VAE and AR-VAE models, utilizing aligned sequence input and raw sequence input respectively, for the generation of functional protein variants. ProteinGAN (Repecka et al., 2021) leverages GANs to learn natural protein sequence distribution. Recently, language models have shown powerful capabilities in NLP, including modeling large-scale data, exceptional transfer learning (Peng et al., 2019), and robust representation (Devlin et al., 2018). These attributes enable their extension to the

analysis of biological sequences (Elnaggar et al., 2021; Rives et al., 2021). Simultaneously, language models also demonstrate their prowess in protein generation. ProtGPT (Ferruz et al., 2022), a ProtLM with a transformer decoder architecture trained on the UniRef50 dataset, is capable of generating protein sequences that are comparable to natural proteins. ProGen (Madani et al., 2020) can generate artificial lysozymes that exhibit catalytic efficiencies similar to those of natural ones through the use of a conditional language model.

## 3 METHODOLOGIES

### 3.1 AUTOREGRESSIVE LANGUAGE MODEL

The autoregressive language model produces the next token based on its previous context in the sequence. Let $a = (a_1, ..., a_L)$ be a protein sequence with $L$ amino acids. The chain rule of probability can be formulated as $p(x) = \prod_{i=1}^{L} p(a_i | a_{<i})$ (Bengio et al., 2000). An autoregressive ProtLM is to learn the distribution of amino acids in proteins, called linkage disequilibrium or evolutionary patterns in Proteomics. In practically, a ProtLM with parameters $\theta$ can be trained by minimizing the negative log-likelihood loss value across a given protein dataset $D = \{a^1, ..., a^{|D|}\}$. The loss function can be formulated as $L = \sum_{k=1}^{|D|} \log p_\theta(a_i^k | a_{<i}^k)$.

Here, we employed a pre-trained language model ProtGPT2 (Ferruz et al., 2022) as the base model for protein generation. ProtGPT2 is a transformer decoder-only language model trained on UniRef50. Its architecture is similar to the previously released GPT2-large Transformer (Radford et al., 2019), which is constructed with a 36-layer transformer decoder and 12 attention heads in each layer with a model dimensionality of 1280.

### 3.2 PREFIX TUNING

Prefix tuning trains a virtual token as a prompt to guide ProtLM in generating proteins with desired functions and structures. The prefix token is a virtual token because it does not represent specific amino acids but is learned through training on a specialized dataset. The prefix token $p$ is concatenated on the left of the protein sequence as the first context, thereby influencing the generation of amino acids on its right. Let $a' = [p; a]$ denote the concatenated input, $p_i$ and $|p|$ be the $i$-th virtual prefix token and length of prefix respectively. The prefix embedding is generated by the prefix encoder $P_\beta \in R^{|p| \times d}$. The output $h_i$ of the ProtLM with a prefix can be formulated as follows:

$$h_i = \begin{cases} P_\beta(i), \text{if } i < |P| \\ ProtLM_\sigma(a_i', h_{<i}), otherwise \end{cases} \tag{1}$$

where $\sigma, \beta$ are parameters of the ProtLM and prefix encoder. The objective function of prefix tuning is the same as autoregressive language model. Furthermore, to deal with the unstable optimization of parameters $\beta$, $P_\beta[i, :]$ is reparameterized by a matrix $(P_\beta')$ into $MLP_\beta(P_\beta'[i, :])$ by using feed-forward neural network. During training, the parameters $\sigma$ in the ProtLM are frozen, while only the parameters $\beta$ in the prefix encoder are trainable. Thus, after training only the prefix encoder $P_\beta$ needs to be stored.

### 3.3 PROTEIN SEQUENCE GENERATION

During the generation process, prefix tokens associated with desired structures and functions are first fed into prefix-tuned ProtLM. Then in every generation step, a subsequent amino acid is produced according to its previous tokens. To increase the generative diversity and reduce the repeated amino acids, top-$k$ sampling (Radford et al., 2019) with a repetition penalty (Keskar et al., 2019) is employed, which is a sampling strategy that selects the subsequent tokens from the top $k$ most probable tokens featured within the distribution generated by the ProtLM. Typically, the top $k$ tokens are sampled by using the softmax function with a sampling temperature. The probability of the $i$-th token being sampled from the top $k$ tokens is computed by Eq. (2).

$$p_i = \frac{\exp(z_i/T)}{\sum_j \exp(z_j/T)} \tag{2}$$

where $z_i$ is the prediction for token $i$, $T$ is the temperature, and $p_i$ is the sampling probability of token $i$. The next token is then chosen by sampling through a multinomial distribution with probabilities $p_i$ clipped at the top-$k$ tokens. To prevent repetition, a repetition penalty is used in top-$k$ sampling to reduce the probabilities of tokens that have been presented within priors. The probability of the $i$-th token is defined as Eq.(3)

$$p_i = \frac{\exp(z_i/T \cdot I(i \in g))}{\sum_j \exp(z_j/T \cdot I(j \in g))}, \ I(i \in g) = \theta \text{ if i} \in \text{g is True else } 1 \tag{3}$$

where $g$ represents a list of generated tokens.

## 4 EXPERIMENTS

### 4.1 DATASETS

We constructed two protein datasets, alpha-helix structure dataset and antimicrobial function dataset, to assess the effectiveness of prefix tuning in directing ProtGPT2 from structure and function aspects. The alpha-helix structure dataset was collected from the Alpha class of Structural Classification of Proteins (SCOP) database (Andreeva et al., 2014). After eliminating duplication, non-protein sequences, and sequences with non-standard amino acids, the final dataset contains 9,168 proteins. The antimicrobial function dataset was collected from the Database of Antimicrobial Activity and Structure of Peptides (DBAASP) (Pirtskhalava et al., 2021; Capecchi et al., 2021). Antimicrobial peptides (AMPs) were classified as active if their IC50 against *P. aeruginosa*, *A. baumanii*, or *S. aureus* are at least one below 10 $\mu$M, 10,000 nM, or 32 $\mu$gmL$^{-1}$. We got a total of 4,505 active peptides. In both experiments, 80% of the data is randomly sampled as training data, and the remaining 20% is the testing data.

### 4.2 METRICS

We measured the quality of generated protein sequences from structure and function aspects by utilizing several metrics, including perplexity, percentage of alpha-helix, Rosetta energy score, probability of antimicrobial function, as well as the identity and diversity.

Perplexity is a general metric in sequence generation tasks, which is the exponential value of the cross-entropy loss computed for each token within a given dataset. A lower perplexity correlates with a higher quality model. The percentage of alpha-helix was calculated by employing the DSSP tool to count the amino acids with alpha-helix structure in the protein structure predicted by ESM-Fold (Lin et al., 2023). We got Rosetta energy scores by running Rosetta Relax with the default Rosetta full-atom energy terms and weights in the Pyrosetta (Chaudhury et al., 2010). Lower Rosetta energy conformers correlate with more relaxed structures (Sauer et al., 2020). To minimize the time needed for relax, we set the maximum number of iterations to 20. A general guideline is that the total score (Rosetta Energy Units, REU) should typically fall within the range of -1 to -3 per residue (Wedemeyer et al., 2019). We calculated the physicochemical properties of generated AMPs, including length, charge, charge density, hydrophobicity, hydrophobic moment, and isoelectric point by using modlAMP(Müller et al., 2017). The antimicrobial function of generated AMPs were predicted by leveraging the CAMP server (Waghu et al., 2014). Sequences with a length greater than 100 amino acids were removed before prediction. The sequence identity was performed by BLAST.

### 4.3 SETUP

We employed the pre-trained ProtGPT2 as the base model, and optimized it with prefix tuning and fine tuning methods on alpha-helix structure dataset and antimicrobial function dataset. To estimate the effectiveness of prefix tuning, we compared the proteins generated by prefix-tuned ProtGPT2 with the proteins generated by pre-trained ProtGPT2 and fine-tuned ProtGPT2, as well as two baselines: the natural proteins in testing data and the randomly generated proteins.

All compared methods were trained in 100 epochs with the AdamW optimizer (Loshchilov & Hutter, 2017). We set a learning rate of 5e-6 and 5e-3 for fine-tuning and prefix tuning with batch size 8 on the alpha-helix structure dataset, and a learning rate of 1e-6 and 1e-3 for fine-tuning and prefix tuning with batch size 32 on antimicrobial function dataset, respectively. We implemented a polynomial

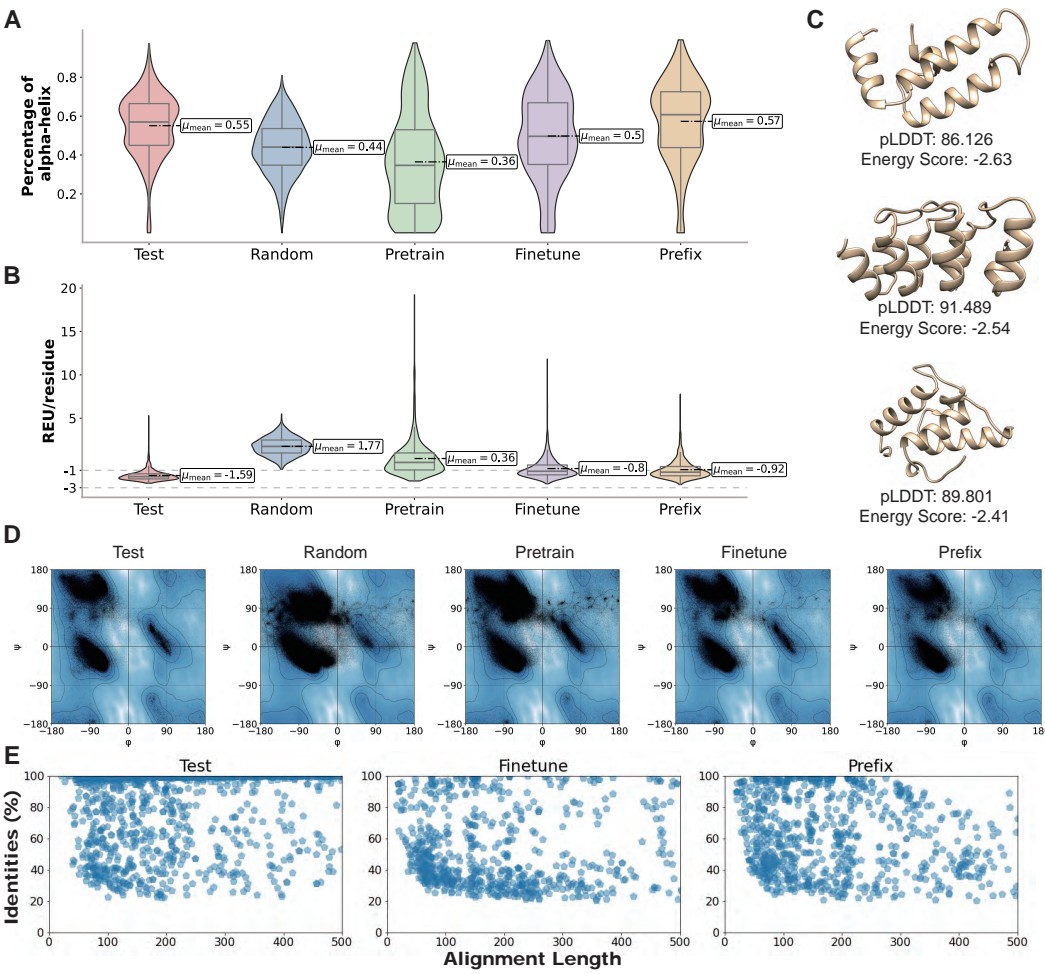

Figure 2: Comparison in generating proteins with alpha-helix structure. (A) Percentage of amino acids within alpha-helix structure in the compared datasets. (B) Average Rosetta energy units per residue in the compared datasets. (C) Three proteins with 'all alpha-helix' generated by prefix-tuned ProtGPT2. (D) Ramachandran plots comparing the $(\phi, \psi)$ dihedral angles for the testing set and generated proteins by different methods. (E) Pairwise sequence identity comparison of testing dataset, proteins generated by fine-tuned and prefix-tuned ProtGPT2 against training dataset.

decay schedule with a final learning rate of 1e-7 and a polynomial exponent of 3. The length of prefix virtual tokens is 100 and 20 for the alpha-helix structure dataset and antimicrobial function dataset, respectively. During generation, the subsequent amino acids were sampled from the top 500 with repetition penalty of 1.2. The random dataset was created by randomly combining 20 amino acids according to their background distribution in the training dataset. We generated 2,000 proteins with alpha-helix structure and 1000 proteins with antimicrobial function by each compared method.

## 5  RESULT

### 5.1  GENERATING PROTEINS WITH ALPHA-HELIX STRUCTURE

In this section, we evaluated the effectiveness of prefix tuning to prompt ProtGPT2 from the alpha-helix structure aspect. We compared the prefix-tuned ProtGPT2 with the pre-trained ProtGPT2, fine-tuned ProtGPT2, as well as random generation according to amino acid background distribution.

We first compared the percentage of amino acids with alpha-helical structure and the Rosetta energy score (Rosetta Energy Units, REU) on the generated proteins by all compared methods against the testing dataset. Prefix-tuned ProtGPT2 outperforms other methods in terms of both metrics. For the alpha-helix structures comparison (**Figure 2A**), although both prefix-tuning and fine-tuning methods significantly improve the percentage of alpha-helix structure in generated proteins comparing to pre-trained ProtGPT2, the prefix tuning achieves closer distribution to natural proteins in the testing dataset, having a higher proportion of amino acids with alpha-helix structures. To further estimate the model's capability to efficiently generate stable proteins, we calculated the Rosetta energy of generated proteins (**Figure 2B**). The energy scores of majority proteins are between -1 and -3 REU/residue, except proteins in the random generation dataset. The mean energy scores for testing, prefix-tuned, fine-tuned datasets are -1.59, -0.92, and -0.8 REU/residue, respectively. The proteins generated by prefix-tuned ProtGPT2 have lower Rosetta energy scores than the proteins generated by the other methods, but are still slightly higher than the natural proteins in the testing dataset. The three 'all alpha' protein examples (**Figure 2C**) generated by prefix tuning also show the stability with low energy scores.

To further analyze the conformation and dihedral angular distribution of proteins, we conducted Ramachandran plots (Ramachandran & Sasisekharan, 1968) to illustrate the co-occurrence frequency of dihedrals ($\phi$, $\psi$). The result (**Figure 2D**) indicates that all methods are able to generate all three major secondary structure elements in protein backbones. However, the proteins generated by prefix tuning exhibit greater similarity to the natural sequence conformation, while other methods produce dihedral angles that are more dispersed and less reasonable, scattered across the periphery. We also conducted a comparison of sequence identity for generated proteins against the training dataset (**Figure2E**). The proteins produced through prefix tuning exhibit a higher sequence identity, whereas the proteins generated by fine-tuning predominantly display identities falling within the range of 20% to 50%. Note that the proteins generated by pre-trained ProtGPT2 and random generation have few sequences matched with training data. Thus their results are not presented.

Overall, prefix tuning with a limited number of trainable parameters surpasses fine tuning with fully trainable parameters in generating proteins with stable energy and reasonable conformation, demonstrating the capability of prefix tuning to prompt large ProtLM with desired structures.

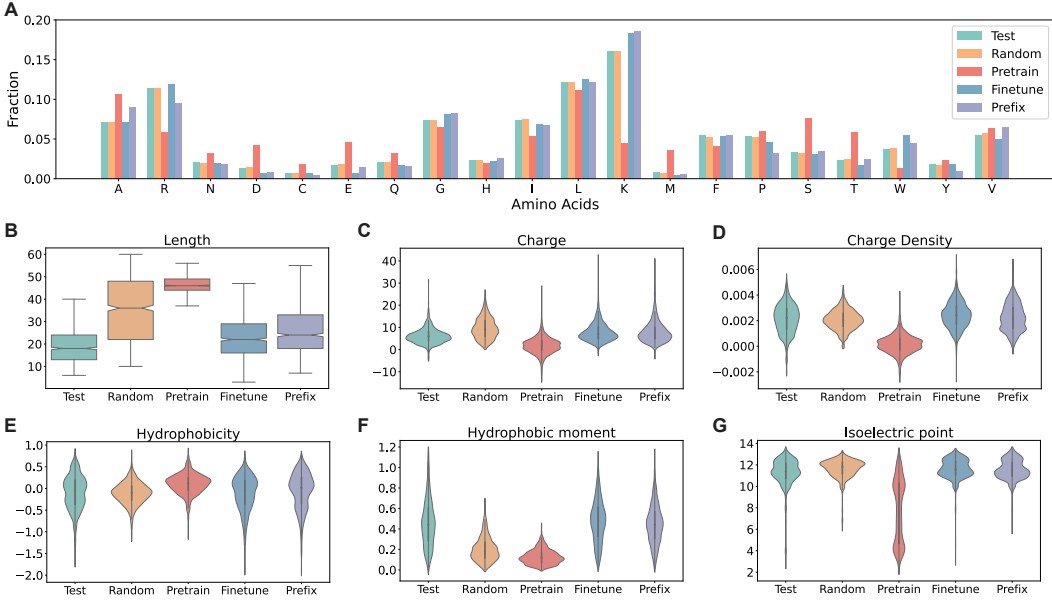

Figure 3: Physicochemical property comparison in generating proteins with antimicrobial function.

## 5.2 GENERATING PROTEINS WITH ANTIMICROBIAL FUNCTIONS

In this section, we evaluated the capability of prefix tuning from the antimicrobial function aspect. We first compared the generated proteins in terms of 7 physicochemical properties, including amino acid distribution, length, charge, charge density, hydrophobicity, hydrophobic moment, and isoelectric point (**Figure 3**). Since the ProtGPT2 was pre-trained on a large-scale protein dataset, which includes peptides with short amino acid sequences and large proteins with long amino acid sequences, we limited the maximum length to 20 amino acids in the generation process to focus on peptide generation. Except for AMPs generated by pre-trained ProtGPT2, all other methods are able to produce AMPs with amino acid distribution close to natural AMPs in the testing dataset (**Figure 3**A). Note that the random generation is based on the background distribution of amino acids in the testing dataset. Thus it has the same distribution with test data. The other results of 6 physicochemical properties distribution show that the AMPs generated by prefix tuning and fine tuning are both notable resemblance in physicochemical properties of natural AMPs in the testing dataset, demonstrating that both of them possess the potential to direct ProtGPT2 toward the generation of protein sequences with target physicochemical properties (Fjell et al., 2012).

We then compared antimicrobial activity of generated AMPs between prefix tuning and several state-of-the-art AMP generation methods, including PepLSTM (Muller et al., 2018), PepCVAE (Das et al., 2018), and Hydramp (Szymczak et al., 2023). PepLSTM designs AMPs by leveraging Long Short-Term Memory (LSTM) to recognize the grammar of amphipathicity in peptides. PepCVAE designs novel AMPs based on a semi-supervised variational autoencoder model. HydrAMP employed a conditional variational autoencoder to project amino acid sequences into reduced-dimensional, continuous representations. The activity of all generated AMPs was predicted by CAMP (Waghu et al., 2014), an open-access AMP prediction web-server (**Figure 4A**). Note that although CAMP has a state-of-the-art performance in AMP identification, it still faces the generalization problem. Thus, the proteins in testing dataset are not predicted as all AMPs.

It is obvious that the prefix-tuned ProtGPT2 produces AMPs with higher probability of being active than the fine-tuned ProtGPT2. And the activity distributions of AMPs generated by prefix-tuned ProtGPT2 and fine-tuned ProtGPT2 are significantly higher than other methods. Similarly, we also conducted a comparison of sequence identity for generated AMPs against the training dataset (**Figure 4B**). AMPs produced by prefix tuning exhibit similar distribution to natural AMPs, demonstrating they have higher sequence identity. In contrast, only one sequence generated by fine-tuning can be matched to the training set. Furthermore, the amphipathicity has been demonstrated to be a significant property influencing the antimicrobial activity of antimicrobial peptides (Fjell et al., 2012). We generated helical wheel plots of 5 top-ranking AMPs with P(AMP)=1 generated by the prefix-tuned ProtGPT2 (**Figure 4C**). The plots show that the generated AMPs have the formation of amphipathic helical structures. The results suggest that prefix tuning can prompt ProtGPT2 to generate proteins with desired functions.

## 5.3 COMBING PREFIX TOKENS FOR FINE-GRAINED CONTROL IN PROTEIN GENERATION

A large language model can be prompted by a complicated instruction that contains multiple tokens for fine-grained control, such as using multiple tokens to specify different aspects or conditions within the instruction. To further investigate the potential of prefix tuning in protein generation, we combined alpha-helix prefix token and AMP prefix token to prompt the ProtGPT2 to generate proteins with both alpha-helix structure and AMP function.

We implemented two manners for combining the alpha-helix prefix token and AMP prefix token: concatenating or averaging their prefix embeddings (**Figure 5**). Regarding the limitation of input length and robustness of token combination, we retrained the alpha-helix prefix embedding with length of 20. To illustrate the impact of the AMP prefix token, we integrated the AMP prefix token into the alpha-helix prefix token, and conducted an ablation experiment by replacing the AMP prefix token with randomly initial prefix tokens. Since the AMPs are a class of small proteins with short amino acid sequences, we filtered out the generated sequences with length exceeding 100 amino acids. After filtering, the peptides prompted by only alpha-helix prefix token are left with 254 sequences, with most of them having length around 80 amino acids and the average activity probability of 0.425. When combining the AMP prefix token into the alpha-helix prefix token, especially using the averaging manner, the probability distribution of being AMP is notably improved and the length

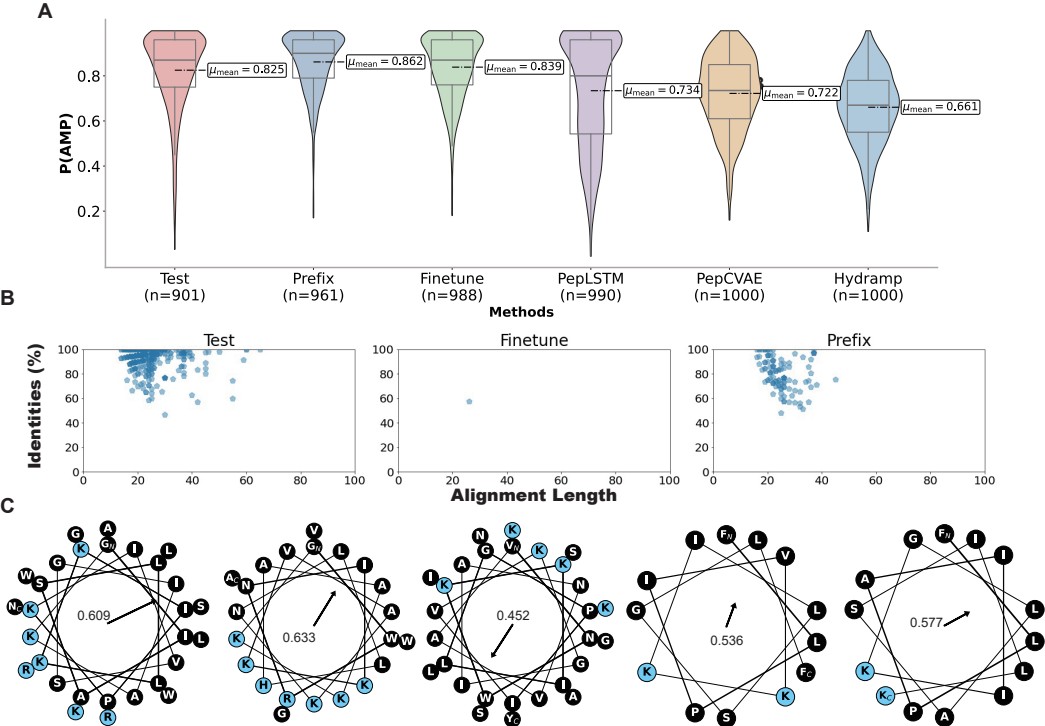

Figure 4: Antimicrobial activity comparison in generating proteins with antimicrobial functions. (A) The probability distribution comparison of being the active AMPs. (B) Pairwise sequence identity comparison of testing dataset, generated by fine-tuned and prefix-tuned ProtGPT2 against training dataset. (C) The helical wheel plots of 5 top-ranking AMps with P(AMP)=1 generated by the prefix-tuned ProtGPT2. The amino acids labeled with "N" and "C" subscript represent the N-terminus and C-terminus of the peptide sequences, respectively. The black circles represent hydrophobic residues, while the blue circles represent polar residues. The spatial segregation between hydrophobic and polar amino acids can be precisely assessed using the hydrophobic moment (HM) and the associated HM vector points toward the hydrophobic face of the helix.

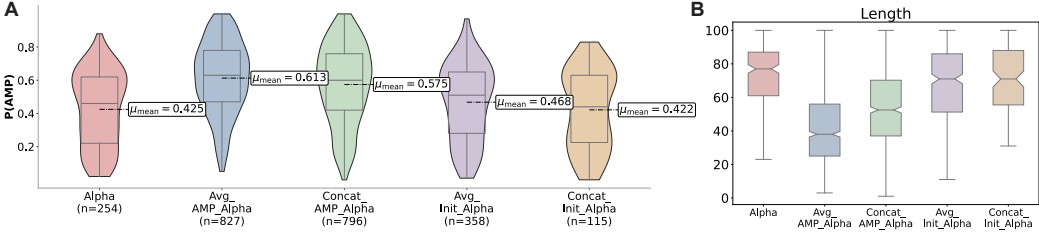

Figure 5: Comparison of probability of antimicrobial activity (A) and sequence length (B) in generating proteins with antimicrobial functions by using prefix token combinations.

of generated peptides becomes shorter. However, the randomly initial prefix token has no discernible impact on the peptide generation. These results demonstrate that combining multiple prefix tokens can effectively fine-grain the protein generation by specifying different properties or conditions. Taking this strength, we can construct more prefix tokens to guide the ProtLM's behavior in a more precise manner, leading to more accurate and desired outputs.

## 5.4 IMPACT OF LOW TRAINING DATA AND LENGTH OF PREFIX EMBEDDING

Prefix tuning has shown significant capability in the low-data setting in the field of NLP. To evaluate the effectiveness of prefix tuning in protein generation tasks in low-data scenarios, we randomly

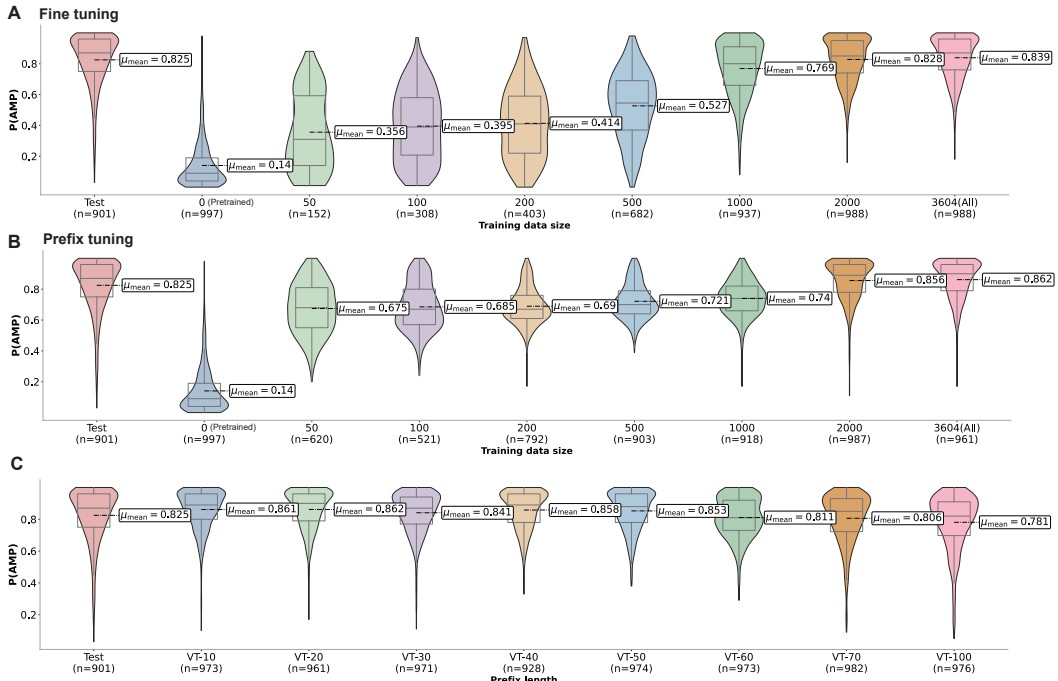

Figure 6: Comparison of antimicrobial activity in low-data settings and various prefix length settings. The probability distribution for AMPs generated by the fine-tuned ProtGPT2 (A) and the prefix-tuned ProtGPT2 (B) in a low-data setting. (C) The probability distribution for AMPs generated by the prefix-tuned ProtGPT2 with various prefix length settings.

subsampled the full antimicrobial function dataset to obtain 6 small datasets with 50, 100, 200, 500, 1000, 2000 AMPs. We compare fine-tuned ProtGPT2 and prefix-tuned ProtGPT2 in a low-data setting (**Figure 6A, B**). Although both of them exhibit an improvement trend in terms of activity probability of generated AMPs as the size of the training data increases, the prefix tuning consistently outperforms the fine-tuning in all instances of low-data settings, especially on the 4 lower-data settings with 50, 100, 200, 500 AMPs. But the gap narrows as the dataset size increases. These results highlight the effectiveness of prefix tuning when dealing with small datasets.

A longer prefix embedding results in more trainable parameters, leading to increased expressive power in the prompt generation, but more parameters also need larger data to training. We evaluated the prefix tuning with different prefix embedding length settings (**Figure 6C**). The performance reaches the peak at a threshold of 20 in the AMP design task and then slightly drops as the length increases. The prefix length is indeed a crucial hyperparameter in prefix tuning.

## 6 CONCLUSION

In this study, we leveraged the prefix tuning to efficiently prompt large protein language models for generating proteins with desired properties. We evaluated the effectiveness of prefix tuning on generating proteins with alpha-helix structure and with antimicrobial function. The results indicate prefix tuning is efficient than fine tuning in terms of higher performance and fewer trainable parameters. Moreover, multiple prefix tokens can be combined to guide the protein language model's behavior in a more precise manner, leading to more accurate and desired outputs.

In future study, we plan to integrate multiple modal information including structure-related representation and function-related features for protein generation. These methods can be effectively extended to various applications in the field of antibody and drug peptide discovery, thereby facilitating advancements in drug discovery research.

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
