data capabilities, exceptional transfer learning abilities (Peng et al., 2019), and robust representation capabilities (Devlin et al., 2018). These attributes enable their extension to the analysis of biological sequences (Elnaggar et al., 2021; Rives et al., 2021). Simultaneously, language models also demonstrate their prowess in protein generation. ProtGPT (Ferruz et al., 2022), a ProtLM with a transformer decoder architecture trained on the UniRef50 dataset, is capable of generating protein sequences with amino acid compositions and disorder propensities that are comparable to natural proteins. ProGen (Madani et al., 2020) can generate artificial lysozymes that exhibit catalytic efficiencies similar to those of natural ones through the use of a conditional language model.

## 3 METHODOLOGIES

### 3.1 AUTOREGRESSIVE LANGUAGE MODEL

Let $a = (a_1, ..., a_L)$ be a protein sequence with $L$ amino acids. The autoregressive language model produces the next token based on its previous context in the sequence. The chain rule of probability can be formulated as (Bengio et al., 2000) $p(x) = \prod_{i=1}^{L} p(a_i|a_{<i})$. An autoregressive ProtLM is to learn the distribution of amino acids in protein sequences, which also called linkage disequilibrium or evolutionary patterns in Proteomics. In practically, a ProtLM with parameters $\theta$ can be trained by minimizing the negative log-likelihood loss value across a given protein dataset $D = \{a^1, ..., a^{|D|}\}$. The loss function can be formulated as follows $L = \sum_{k=1}^{|D|} \log p_\theta(a_i^k|a_{<i}^k)$.

Here, we employed a pre-trained language model ProtGPT2 (Ferruz et al., 2022) as the base model for protein generation. ProtGPT2 is a transformer decoder-only language model trained on UniRef50. Its architecture is similar to the previously released GPT2-large Transformer (Radford et al., 2019), which is constructed with a 36-layer transformer decoder and 12 attention heads in each layer with a model dimensionality of 1280.

### 3.2 PREFIX TUNING

Prefix tuning trains a virtual token as a prompt to guide the ProtLM in generating protein sequences with desired functions and properties. The prefix token is a virtual token because it does not represent specific or unique amino acids but is learned through training on a specialized dataset. The virtual prefix token $p$ is concatenated on the left of the protein sequence as the first context, thereby influencing the generation of following amino acids on its right. Let $a' = [p; a]$ denote the concatenated input, $p_i$ and $|p|$ be the $i$-th virtual prefix token and length of prefix respectively. The prefix embedding is generated by the prefix encoder $P_\beta \in R^{|p| \times d}$. The output $h_i$ of the ProtLM with a prefix can be formulated as follows:

$$h_i = \begin{cases} P_\beta(i), \text{if i} < |\text{P}| \\ LM_\sigma(a_i', h_{<i}), otherwise \end{cases} \tag{1}$$

where $\sigma, \beta$ are trainable parameters of the ProtLM and prefix encoder.

The objective function of prefix tuning is the same as autoregressive language model. During training, the parameters $\sigma$ in the ProtLM are frozen, while only the parameters $\beta$ in the prefix encoder are trainable. Furthermore, to deal with the unstable problem of parameters $\beta$ optimization, $P_\beta[i, :]$ will be reparameterized by a matrix $(P_\beta')$ into $MLP_\beta(P_\beta'[i, :])$. Since parameters $\sigma$ in the ProtLM are not changed, after training only the prefix encoder $P_\beta$ needs to be stored.

### 3.3 PROTEIN SEQUENCE GENERATION

During the generation process, prefix tokens associated with desired structures and functions are first fed into prefix-tuned ProtLM. Then in every generation step, a subsequent amino acid is produced according to its previous tokens. To increase the generative diversity and reduce the repeated amino acids, top-$k$ sampling (Radford et al., 2019) with a repetition penalty (Keskar et al., 2019) is employed, which is a sampling strategy that selects the subsequent tokens from the top $k$ most probable tokens featured within the distribution generated by the ProtLM. Typically, the top $k$ tokens are sampled by using the softmax function with a sampling temperature. The probability of the $i$-th

token being sampled from the top $k$ tokens is computed by Eq. (2).

$$p_i = \frac{\exp(z_i/T)}{\sum_j \exp(z_j/T)} \tag{2}$$

where $z_i$ is the prediction for token $i$, $T$ is the temperature, and $p_i$ is the sampling probability of token $i$. The next token is then chosen by sampling through a multinomial distribution with probabilities $p_i$ clipped at the top-$k$ tokens. To prevent repetition, the repetition penalty is used in top-$k$ sampling to reduce the probabilities of tokens that have been presented within priors. The probability of the $i$-th token is defined as Eq.(3)

$$p_i = \frac{\exp(z_i/T \cdot I(i \in g))}{\sum_j \exp(z_j/T \cdot I(j \in g))}, \; I(i \in g) = \theta \text{ if i} \in \text{g is True else 1} \tag{3}$$

where $g$ represents a list of generated tokens.

## 4 EXPERIMENTS

### 4.1 DATASETS

We constructed two protein datasets, alpha-helix structure dataset and anti-microbial function dataset, to assess the effectiveness of prefix tuning in directing ProtGPT2 to generate proteins with specific structure and function. The alpha-helix structure dataset was collected from the Alpha class of Structural Classification of Proteins (SCOP) database (Andreeva et al., 2014). After eliminating duplication, non-protein sequences, and sequences with non-standard amino acids, the final dataset contains 9168 proteins sequences. The anti-microbial function dataset was collected from the Database of Antimicrobial Activity and Structure of Peptides (DBAASP) (Pirtskhalava et al., 2021; Capecchi et al., 2021). Anti-microbial peptides (AMPs) were classified as active if their IC50 against *P. aeruginosa*, *A. baumannii*, or *S. aureus* are at least one below 10 $\mu$M, 10,000 nM, or 32 $\mu$gmL$^{-1}$. We got a total of 4,505 active peptides. In both experiments, 80% of the data is randomly sampled as training data, and the remaining 20% is the testing data.

### 4.2 METRICS

We measured the quality of generated protein sequences from structure and function aspects by utilizing several metrics.

**Perplexity**. Perplexity is a general metric in sequence generation tasks, which is the exponential value of the cross-entropy loss computed for each token within a given dataset. We use perplexity to evaluate the quality of prefix-tuned ProtGPT2 in training. A lower perplexity correlates with a higher quality model.

**Structure prediction**. We predicted the protein structure by using ESMFold (Lin et al., 2023), a large language model-based protein structure predictor that can direct inference of full atomic-level protein structure from protein sequence. ESMFold is founded upon a BERT-base language model named ESM2, which encompasses a vast parameter count of 15 billion and is trained by masked language modeling objective (Devlin et al., 2018) on the UniRef50 dataset. After getting the protein structure, we employed the DSSP tool to conduct an in-depth analysis of protein secondary structure and each amino acid is labeled with the corresponding secondary structure property including $\alpha$-helix(H), Isolated $\beta$-bridge residue(B), Strand(E), 3-10 helix(G), $\pi$-helix(I), Turn(T), Bend(S) and other(-).

**Rosetta calculations**. Rosetta Relax utilizes the Rosetta energy function as a standard and employs the Monte Carlo optimization method to optimize the protein conformation. After threading the amino acid sequence through the known structure, energy minimization/relaxation processes are executed. These procedures enable the backbone to potentially transition to a lower energy state (Madani et al., 2020). As a result of this process, diverse conformations of both the backbone and rotamer configurations are generated. Lower Rosetta Energy conformers correlate with more relaxed structures (Sauer et al., 2020). Rosetta Relax runs in the Pyrosetta(Chaudhury et al., 2010) with predicted structure by ESMFold. We use the energy score function with the default Rosetta full-atom energy terms and weights. To minimize the time needed for relax, we set the maximum

number of iterations to 20. A general guideline is that the total score (Rosetta Energy Units, REU) should typically fall within the range of -1 to -3 per residue (Wedemeyer et al., 2019).

**Antimicrobial analysis**. We calculate the physicochemical properties of generated antimicrobial peptides (AMPs) including length, charge, charge density, hydrophobicity, hydrophobic moment, and isoelectric point by using modlAMP(Müller et al., 2017). We also assessed the antimicrobial function of generated AMP sequences by leveraging the CAMP server (Waghu et al., 2014), which facilitates the construction of prediction models, including Support Vector Machines (SVM), Random Forest (RF), and Artificial Neural Networks (ANN). We selected the RF model that demonstrated the highest level of performance. During the prediction process, sequences with a length greater than 100 are removed. This evaluation was conducted to gauge the antimicrobial potential of the generated sequences and their propensity for exhibiting antimicrobial activity.

## 4.3    SETUP

The proposed models were trained in 100 epochs with batch size 8 in the structure special dataset and batch size 32 in the function special dataset. We employed the AdamW optimizer (Loshchilov & Hutter, 2017). Additionally, we implemented a polynomial decay schedule, utilizing specific parameters such as a final learning rate of 1e-7 and a polynomial exponent of 3. Because of the distinction between parameter adjustments resulting from fine-tuning and the re-training of virtual token embeddings inherent to prefix tuning. We designated a learning rate of 5e-6 for fine-tuning and a learning rate of 5e-3 for prefix tuning on the structure special dataset. We also set a learning rate of 1e-6 and 1e-3 for fine-tuning and prefix tuning respectively on function special dataset. To effectively tune the generation of amino acid sequence on diverse dataset types, we also set different virtual token length. Specifically, the model on structure-specific dataset incorporates a virtual token length of 100, while the model on function-specific dataset uses a virtual token length of 20.

We generate new amino acids by sampling from top 500 with repetition penalty of 1.2 in all settings. In the structure-special setting, we generate a total of 2,000 sequences. In the function-special setting, a total of 1000 sequences are generated. The random dataset was created by randomly combining 20 amino acids according to the background distribution in the training dataset. The pretrain dataset was synthesized using ProtGPT2 without any form of tuning while maintaining consistent generation parameters aligned with prefix tuning, except the max length was set to 20 in the function-special setting. The finetune dataset was constructed by ProtGPT2 after fine-tuning.

## 5    RESULT

### 5.1    GENERATING PROTEINS WITH ALPHA-HELIX STRUCTURE

We evaluated the effectiveness of prefix tuning to prompt the ProtGPT2 on generation of proteins with $\alpha$-helix structure. We compared the prefix-tuned ProtGPT2 with the pre-trained ProtGPT2, fine-tuned ProtGPT2, as well as random generation according to amino acid background distribution.

We first compared the percentage of amino acids with $\alpha$-helical secondary structures (%) and the Rosetta energy score (Rosetta Energy Units, REU) on the testing dataset and the generated proteins by all compared methods. Prefix-tuned ProtGPT2 outperforms other methods in terms of both metrics. For the $\alpha$-helical secondary structures comparison (**Figure 2A**), although the fine-tuning method significantly improves the percentage of $\alpha$-helix structure in generated proteins comparing to pre-trained ProtGPT2, the prefix tuning achieves closer distribution to real proteins in the testing dataset, having a higher proportion of amino acids with $\alpha$-helix structures. To further validate the model's capability to efficiently generate stable proteins, as opposed to solely textual patterns, we conducted a comprehensive analysis of the stability of the generated protein structures. For the Rosetta energy score comparison (**Figure 2B**), the energy scores of majority proteins are between -1 and -3 REU/residue, except proteins in the random generation dataset. The mean energy scores for these datasets are -1.59, -0.92, and -0.8 REU/residue, respectively. The generated proteins by prefix-tuned ProtGPT2 have lower Rosetta energy scores than the proteins generated by the other methods, but are still slightly higher than the natural proteins in testing dataset.

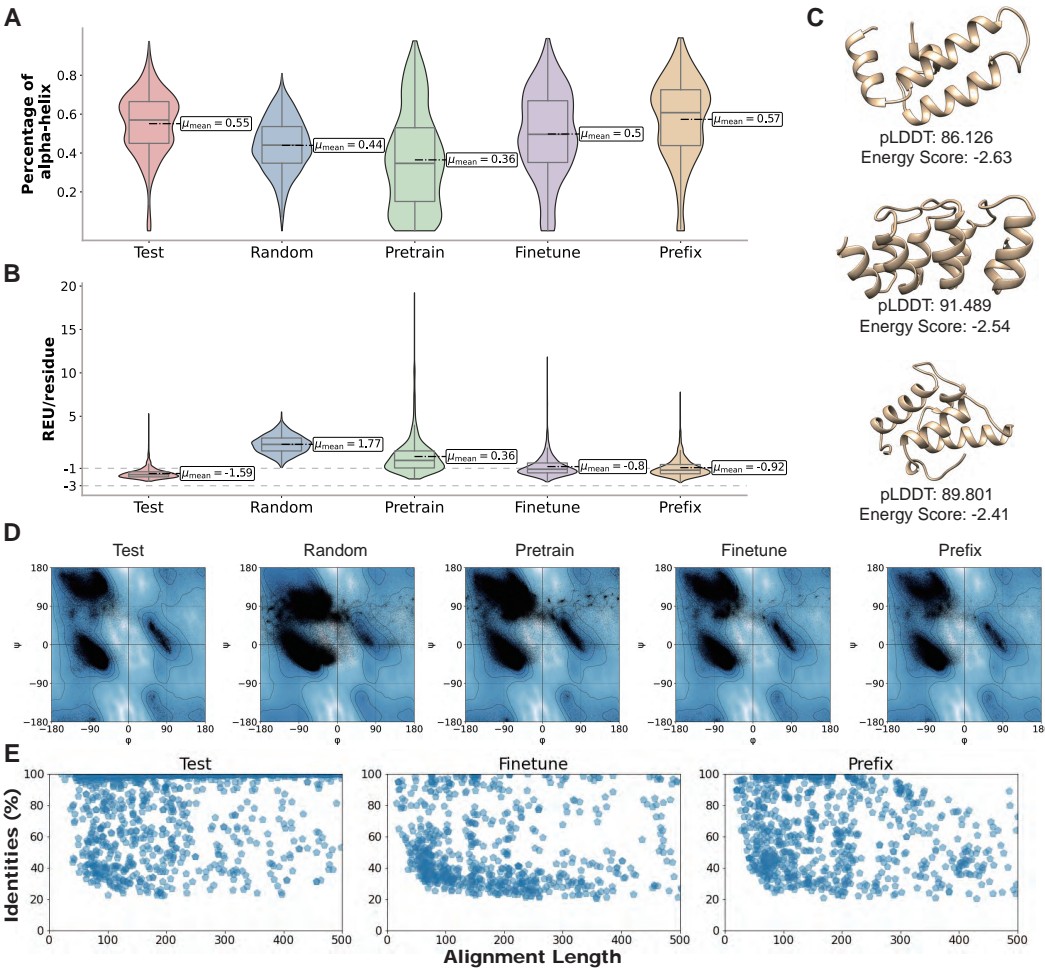

Figure 2: Comparison in generating proteins with alpha-helix structure. (A) The ratio of amino acids within $\alpha$-helical secondary structures for the generated datasets by different methods. (B) Average Rosetta energy units per residue for the generated datasets by different methods. (C) shows three examples of 'all alpha protein' generated by prefix tuning model. (D) Ramachandran plots comparing the $(\phi, \psi)$ dihedral angles for the test set and generated protein sequence by different methods. (E) Comparative analysis of pairwise sequence identities versus alignment length for generated sequences.

To further analyze the conformation and dihedral angular distribution of proteins, we conducted Ramachandran plot (Ramachandran & Sasisekharan, 1968) experiments to illustrate the co-occurrence frequency of dihedrals $(\phi, \psi)$. The result (**Figure 2D**) indicates that all methods are able to generate all three major secondary structure elements in protein backbones. However, the protein sequences generated using the prefix tuning exhibit greater similarity to the natural sequence conformation, while other methods produce dihedral angles that are more dispersed and less reasonable, scattered across the periphery. (**Figure 2C**) shows three examples of 'all alpha protein' generated by prefix tuning model. We also conduct a comparative analysis of sequence identities for generated sequences, determined using Blastp against the training database. The sequences produced through prefix tuning exhibit a higher degree of conservation, whereas the sequences generated during fine-tuning predominantly display identities falling within the range of 20% to 50%.

Overall, prefix tuning with a limited number of trainable parameters surpasses fine-tuning with fully trainable parameters in generation of proteins with stable energy and reasonable conformation of special structures. These results demonstrate the capability of prefix tuning to prompt large ProtLM with desired structures.

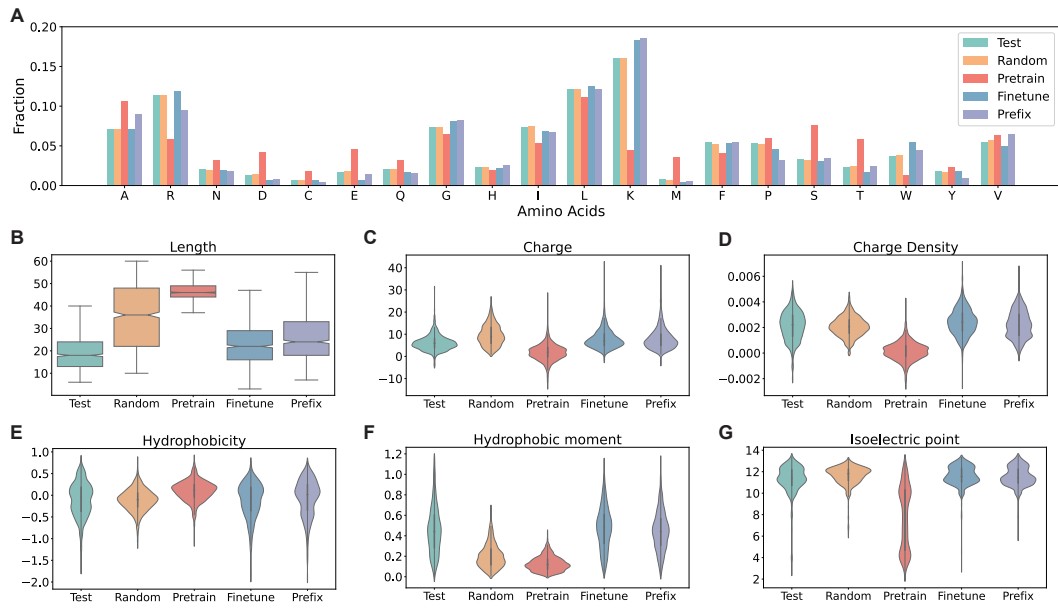

Figure 3: Physicochemical property comparison in generating proteins with antimicrobial function.

## 5.2 GENERATING PROTEINS WITH ANTIMICROBIAL FUNCTIONS

We evaluated the capability of prefix tuning for generating function-specific proteins through AMP design. We first compared prefix tuned ProtGPT2 with fine-tuned ProtGPT2, pre-trained ProtGPT2, random generation in terms of 7 physicochemical properties, including amino acid distribution, length, charge, charge density, hydrophobicity, hydrophobic moment, and isoelectric point (**Figure 3**). Since the ProtGPT2 was pre-trained on a large-scale protein dataset, which includes peptides with short amino acid sequences and large proteins with long amino acid sequences, we limited the maximum length to 20 amino acids in the generation process to focus on peptide generation. Except for AMPs generated by pre-trained ProtGPT2, all other methods are able to produce AMPs with amino acid distribution close to natural AMPs in the testing dataset (**Figure 3**A). Note that the random generation is based on the background distribution of amino acids in testing dataset. Thus it has the same distribution with test data. The other results of 6 physicochemical properties ditribution show that the AMPs generated by prefix tuning and fine tuning are both notable resemblance in physicochemical properties of natural AMPs in the testing dataset, demonstrating that both of them possess the potential capability to direct ProtGPT2 toward the generation of protein sequences with target physicochemical properties (Fjell et al., 2012).

We then compared antimicrobial activity of generated peptides between prefix tuning and several state-of-the-art AMP generation methods, including PepLSTM (Muller et al., 2018), PepCVAE (Das et al., 2018), and Hydramp (Szymczak et al., 2023). PepLSTM designs peptides by leveraging Long Short-Term Memory (LSTM) to recognize the grammar of amphipathicity in peptides. PepCVAE designs novel antimicrobial peptides based on a semi-supervised variational autoencoder model. HydrAMP employed a conditional variational autoencoder to project amino acid sequences into reduced-dimensional, continuous representations. The activity of all generated AMPs was predicted by CAMP (Waghu et al., 2014), an open-access AMP prediction web-server, which outputs probability of being active (Figure 4A). While antimicrobial peptide identifiers are currently prevalent in computational screening methods, they continue to face challenges in terms of their limited generalization capabilities.

It is obvious that the prefix-tuned ProtGPT2 produces peptides with higher probability of being active AMPs than the fine-tuned ProtGPT2. And the activity distributions of AMPs generated by prefix-tuned ProtGPT2 and fine-tuned ProtGPT2 are significantly higher than other methods. In the analysis of sequence identities, parts of sequences produced through prefix tuning exhibit distributional characteristics similar to natural sequences in terms of their identities. In contrast, only one

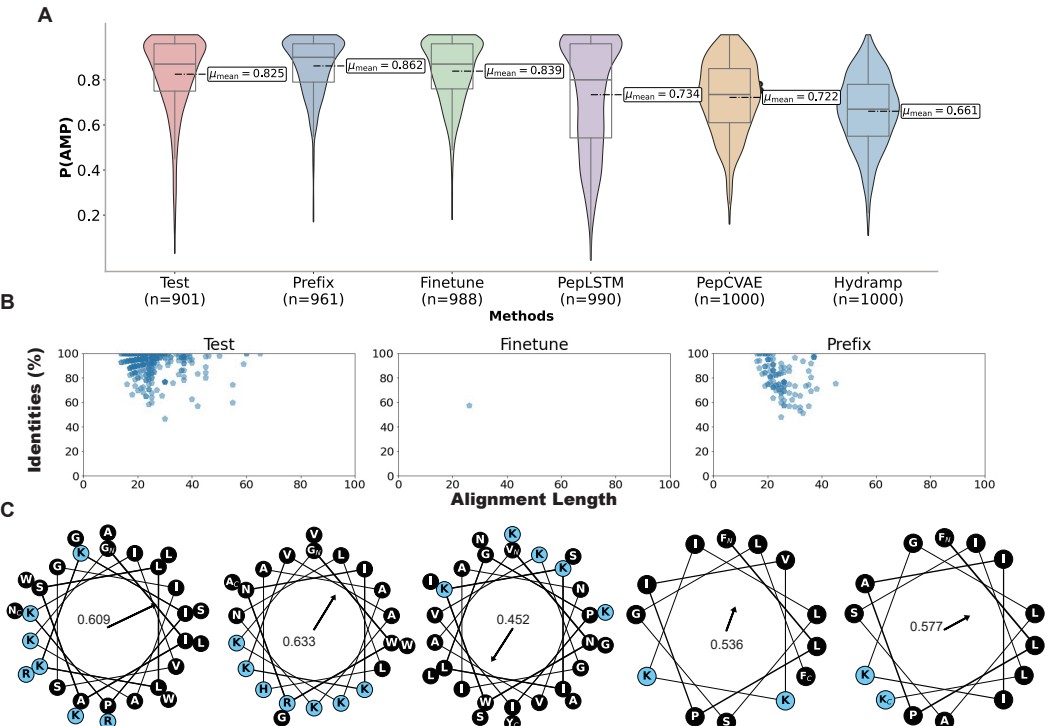

Figure 4: Antimicrobial activity comparison in generating proteins with antimicrobial functions. (A) The active distribution comparison of being active AMPs. P(AMP) represents the probability of sequence to be AMPs. (B) Comparative analysis of pairwise sequence identities versus alignment length for generated sequences. (C) The helical wheel plots of five top-ranking peptides with P(AMP)=1 from the AMPs generated by the prefixtuned ProtGPT2. The amino acids labeled with "N subscript" and "C subscript" represent the N-terminus and C-terminus of the peptide sequences, respectively. The black circles represent hydrophobic residues, while the blue circles represent polar residues. The spatial segregation between hydrophobic and polar amino acids within a protein can be precisely assessed using the hydrophobic moment (HM) and the associated HM vector points toward the hydrophobic face of the helix.

sequence generated during fine-tuning can be effectively compared to the training set. Furthermore, amphipathicity has been demonstrated to be a significant property influencing the antimicrobial activity of antimicrobial peptides (Fjell et al., 2012). We generated helical wheel plots of five top-ranking peptides with P(AMP)=1 from the AMPs generated by the prefix-tuned ProtGPT2 (Figure 4B). The plots show that the generated sequences have the formation of amphipathic helical structures. The results suggest that prefixtuning can prompt ProtGPT2 to generate protein sequences with targeted functions.

### 5.3 COMBING MULTIPLE PREFIX TOKENS FOR FINE-GRAINED CONTROL IN PROTEIN GENERATION

A large language model can be prompted by a complicate instruction that contains multiple tokens for fine-grained control, such as using multiple tokens to specify different aspects or conditions within the instruction. To further investigate the potential of prefix tuning in protein generation, we combined alpha-helix prefix token and AMP prefix token to prompt the ProtGPT2 to generate proteins with both alpha-helix structure and AMP function.

We implemented two methods for combining the alpha-helix prefix token and AMP prefix token by concatenating or averaging their prefix embeddings (**Figure 5**). Regarding to the limitation of input length and robustness of token combination, we retrained the alpha-helix prefix embedding with length of 20. To illustrate the impact of the AMP prefix token, we integrated the AMP prefix

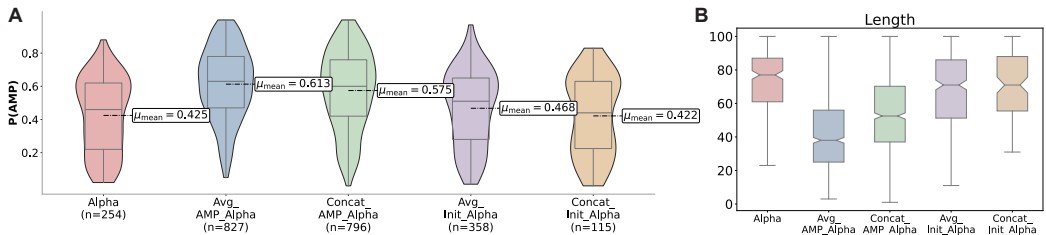

Figure 5: Comparison of probability of antimicrobial activity (A) and sequence length (B) in generating proteins with antimicrobial functions by using multiple prefix token combination.

token into the alpha-helix prefix token, and conducted an ablation experiment by replace the AMP prefix token with randomly initial prefix token. Since the peptides is a class of small proteins with short amino acid sequences, we filtered out the generated sequences with lengths exceeding 100 amino acids. After filtering, the peptides prompted by only alpha-helix prefix token are left 254 sequences, with most of them having length around 80 amino acids and the average probability of 0.425. When combining the AMP prefix token into the alpha-helix prefix token, especially using averaging manner, the probability distribution of being AMP are notably improved and the length of generated peptides become shorter. However, the randomly initial prefix token has no discernible impact on the peptide generation. These results demonstrate that combing multiple prefix tokens can effectively fine-grain the protein generation by specifying different properties or conditions. Taking this strength, we can construct more prefix tokens to guide the ProtLM's behavior in a more precise manner, leading to more accurate and desired outputs.

### 5.4 IMPACT OF LOW TRAINING DATA AND LENGTH OF PREFIX EMBEDDING

Prefix tuning has shown significant capability in the low-data setting in the field of natural language processing. To evaluate the effectiveness of prefix tuning in protein generation tasks in low-data scenarios, we randomly subsampled the full anti-microbial function dataset to obtain 6 small datasets with 50, 100, 200, 500, 1000, 2000 peptides. We compare fine-tuned ProtGPT2 and prefix-tuned ProtGPT2 in low-data setting (**Figure 6**A, B). Although both of them exhibit an improvement trend in terms of probability distribution of generated AMPs as the size of the training data increases, the prefix tuning consistently outperforms the fine-tuning in all instances of low-data settings, especially on the 4 lower-data settings with 50, 100, 200, 500 peptides. But the gap narrows as the dataset size increases. These results highlight the effectiveness of prefix tuning when dealing with small datasets.

A longer prefix embedding results in more trainable parameters, leading to increased expressive power in the prompt generation, but more parameters also need larger data to training. We evaluated the prefix tuning with different prefix embedding length settings (**Figure 6**C). The performance reaches the peak at a threshold of 20 in the AMP design task and then slightly drops as the length increasing. The prefix length is indeed a crucial hyperparameter in prefix tuning.

## 6 CONCLUSION

In this study, we leveraged the prefix tuning to efficiently prompt large protein language models for generating proteins with desired properties. We evaluated the effectiveness of prefix tuning on two protein generation tasks, proteins with alpha-helix structure and proteins with anti-microbial function. The results indicate prefix tuning is more efficient than fine tuning in terms of higher performance and fewer trainable parameters. Moreover, multiple prefix tokens can be combined to guide the protein language model's behavior in a more precise manner, leading to more accurate and desired outputs.

In future study, we plan to integrate multiple modal information including structure-related representation and function-related features for protein generation. These methods can be effectively extended to various applications in the field of antibody and drug peptide discovery, thereby facilitating advancements in drug discovery research.

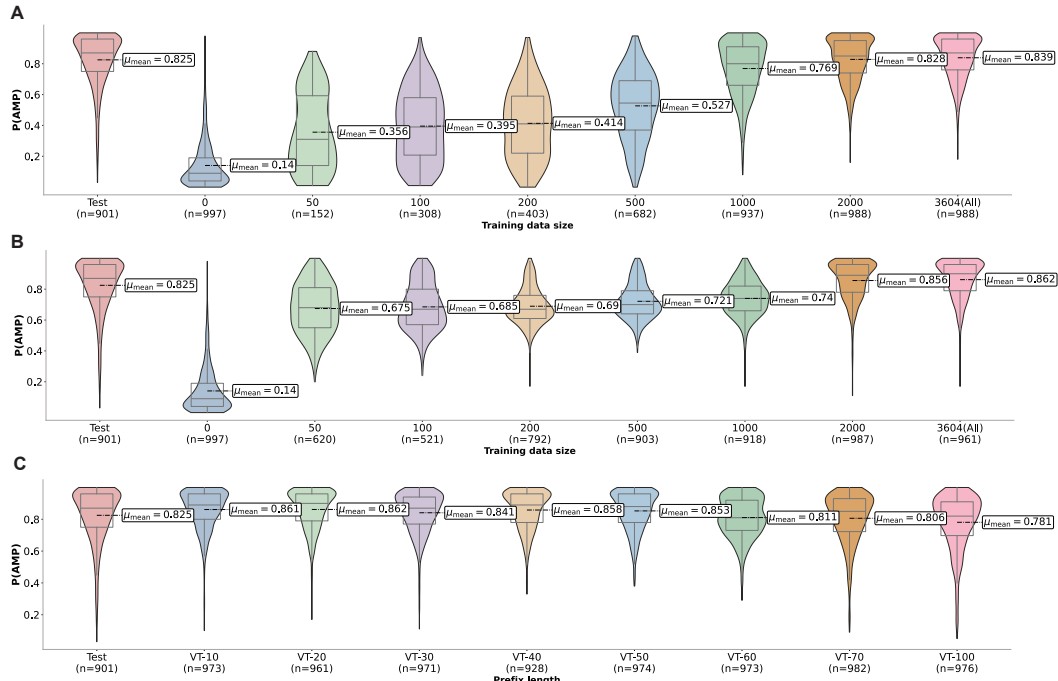

Figure 6: Comparison of antimicrobial activity in low-data settings and various prefix length settings. The probability distribution for AMPs generated by the fine-tuned ProtGPT2 (A) and the prefix-tuned ProtGPT2 (B) in a low-data setting. (C) The probability distribution for AMPs generated by the prefix-tuned ProtGPT2 across various prefix length settings.