# OpenReview forum: "Efficient Parameter Tuning of Large Protein Language Models for De Novo Protein Design"
_ICLR.cc/2024/Conference — ICLR 2024 Conference Withdrawn Submission_

### Official Review · Reviewer_wnAV · 2023-10-26

**Soundness:** 1 poor
**Presentation:** 3 good
**Contribution:** 1 poor
**Rating:** 3
**Confidence:** 4

**Summary:**

This paper studied prefix-tuning, which is one of the popular efficient parameter-efficient tuning methods used in language models fine-tuning, in the realm of Protein language models (ProtLMs). The prefix-tuning is conducted following the same piepline in NLP. The paper studied the prefix-tuning on two different datasets: antimicrobial peptide and alpha-helix structure, as two specific domains for protein generation. The experimental results demonstrate the superiority of prefix-tuning in ProtLMs.

**Strengths:**

- The experimental anaylsis is comprehensive.
- The writing is brilliant. The paper is very easy to follow.

**Weaknesses:**

- The title is overstated as the paper only studied one parameter-efficient tuning paradigm, i.e., prefix-tuning. There are more parameter-efficient tuning methods such as LoRA [1] and Adapter [2]. Therefore, it seems unsuitable to use "Efficient parameter tuning" in the title.

- The analysis why fine-tuning is inferior to prefix-tuning is missing. It is widely accepted that parameter-efficient tuning is usually weaker than fine-tuning, as there are less parameters trained in the learning process. From the theoretical perspective, the expressivity of fine-tuning is stronger than parameter-efficient tuning. However, this paper presented a surprising result that prefix-tuning is consistently better than fine-tuning. The essential explanation on this observation is necessary to make it convincing.

- Many of the experimental details are missing. In section 4.3, the hyper-parameters are set without any explanation, which leads to many questions. For example,

  - Why were all the compared methods trained in 100 epochs? Why don't we perform validation or early stopping?
  - Why was the learning rate of fine-tuning set as 5e-6 and the learning rate of prefix-tuning set as 5e-3? Will it be better if we search from many learning rate candidates?
  - How did you choose the length of prefix virtual token for different datasets?

  The obscurity on hyper-parameter settings will strongly hurt the reliability of the experiments. This also relates to the last weakness I proposed.

- Apart from the surprising conclusion that prefix-tuning is better than fine-tuning, this paper seems to have too little contents. The paper did not propose any new techniques or theorys, and the emprical study is constrained in two datasets and one model.

**Questions:**

No questions for me.

---

### Official Review · Reviewer_H4xA · 2023-11-01

**Soundness:** 2 fair
**Presentation:** 2 fair
**Contribution:** 2 fair
**Rating:** 5
**Confidence:** 4

**Summary:**

The paper introduces and evaluates the application of prefix tuning to optimize protein language models (ProtLMs) for the targeted design of proteins with specific structures and functions.

With prefix tuning, it requires the optimization of fewer parameters than traditional fine-tuning yet yields better results even in situations with limited data on the alpha-helix structure dataset and antimicrobial function dataset. In addition, it demonstrates the ability to merge two prefix virtual tokens, allowing for the generation of proteins with combined desired properties.

The authors benchmark the prefix tuning applied to ProGPT2 with the current methods and through metrics such as perplexity and probability of antimicrobial function, prefix tuning notably surpasses the performance of current methods.

**Strengths:**

1. The application of prefix tuning to protein language models (ProtLMs) represents a creative integration of NLP techniques into the protein engineering domain. And the paper not only applies prefix tuning but also showcases the ability to merge two prefix virtual tokens, which is a fresh perspective in protein sequence generation.
2. By benchmarking prefix tuning on ProtGPT2 against existing methods with comprehensive metrics from structure and function aspects, the authors present a holistic view of its performance, thereby elevating the quality of their results.

**Weaknesses:**

1. While the paper introduces the concept of prefix tuning to protein language models (ProtLMs), it is essentially borrowing a methodology previously established in the NLP domain and applying it directly to ProtGPT2. We cannot deny the value of this approach in drug discovery, but this raises concerns about the true novelty of the approach, as it might be perceived as a straightforward application of an existing technique to a new dataset.

**Questions:**

1. While metrics like perplexity and the probability measure of antimicrobial function provide a quantitative assessment of the generated sequences, how reliably do these metrics correlate with the actual structural integrity, functional viability, and therapeutic efficacy of the proteins when synthesized and tested in real-world lab conditions? In other words, do a lower perplexity and a higher probability of antimicrobial function genuinely translate to a functional protein in a biological context, or might there exist potential discrepancies between the model's predictions and empirical outcomes?

---

### Official Review · Reviewer_xTS7 · 2023-11-04

**Soundness:** 3 good
**Presentation:** 3 good
**Contribution:** 2 fair
**Rating:** 3
**Confidence:** 4

**Summary:**

This paper explore efficient parameter tuning of large protein language models for de novo protein design. The authors introduce the concept of prefix tuning from language modeling research in NLP into protein language models, where only the prefix virtual token is trainable while the pre-trained protein language model is frozen. Two prefix virtual tokens are trained on different datasets, demonstrating the effectiveness of prefix tuning in optimizing fewer trainable parameters and achieving superior results compared to fine tuning. The combination of these two prefix virtual tokens allows for precise control over protein generation with desired properties.

**Strengths:**

(+) The paper clearly demonstrates effectiveness of prefix tuning for protein language models with extensive experiments.

**Weaknesses:**

(-) The major issue lies in the fact that the novelty of the paper is quite limited. The idea of applying prefix tuning from NLP to protein language models does not in itself constitute a significant scientific breakthrough. It would be crucial for the authors to shed light on any new advancements or unique adaptations made specifically for dealing with protein language models, thereby drawing a clear line of difference with existing methods in NLP.

(-) The paper's experiments are solely centered around autoregressive language models for protein sequence generation. However, the scope of protein language models extends beyond this specific type, and includes, for example, masked language models like ESM series give rise to impressive performance for various protein property prediction problem. Testing the applicability of prefix tuning solely on autoregressive models restricts the generalizability of the findings. The paper could significantly benefit from a broader range of experiments to validate how well the concept of prefixing tuning could be applied to various types of protein language models.

**Questions:**

see weaknesses